# Mental Health Conditions– and Substance Use—Associated Emergency Department Visits during the COVID-19 Pandemic in Nevada, USA

**DOI:** 10.3390/ijerph20054389

**Published:** 2023-03-01

**Authors:** Zahra Mojtahedi, Ying Guo, Pearl Kim, Parsa Khawari, Hailey Ephrem, Jay J. Shen

**Affiliations:** 1Department of Healthcare Administration and Policy, School of Public Health, University of Nevada, Las Vegas, NV 89154, USA; 2Department of Environmental and Occupational Health, School of Public Health, University of Nevada, Las Vegas, NV 89154, USA; 3School of Medicine, University of Nevada, Las Vegas, NV 89154, USA

**Keywords:** emergency department, opioids, cannabis

## Abstract

Background—Mental health conditions and substance use are linked. During the COVID-19 pandemic, mental health conditions and substance use increased, while emergency department (ED) visits decreased in the U.S. There is limited information regarding how the pandemic has affected ED visits for patients with mental health conditions and substance use. Objectives—This study examined the changes in ED visits associated with more common and serious mental health conditions (suicidal ideation, suicide attempts, and schizophrenia) and more commonly used substances (opioids, cannabis, alcohol, and cigarettes) in Nevada during the COVID-19 pandemic in 2020 and 2021 compared with the pre-pandemic period. Methods—The Nevada State ED database from 2018 to 2021 was used (*n* = 4,185,416 ED visits). The 10th Revision of the International Classification of Diseases identified suicidal ideation, suicide attempts, schizophrenia, and the use of opioids, cannabis, alcohol, and cigarette smoking. Seven multivariable logistic regression models were developed for each of the conditions after adjusting for age, gender, race/ethnicity, and payer source. The reference year was set as 2018. Results—During both of the pandemic years (2020 and 2021), particularly in 2020, the odds of ED visits associated with suicidal ideation, suicide attempts, schizophrenia, cigarette smoking, and alcohol use were all significantly higher than those in 2018. Conclusions—Our findings indicate the impact of the pandemic on mental health- and substance use-associated ED visits and provide empirical evidence for policymakers to direct and develop decisive public health initiatives aimed at addressing mental health and substance use-associated health service utilization, especially during the early stages of large-scale public health emergencies, such as the COVID-19 pandemic.

## 1. Introduction

Mental health and substance use issues are intertwined, and both reportedly increased in the United States and globally during the COVID-19 pandemic [1,2,3,4,5]. Nevada has distinct characteristics in terms of mental health and substance use, with rates that are typically higher than the national average [6]. Nevada ranks 44th out of 51 states in the USA regarding the prevalence of mental illnesses [7]. Nevada is also one of the worst-performing states for access to mental health care services (39 out of 51 states) [8]. The pandemic had a significant economic impact on Nevada, a world tourism center. More than 90% of those employed in the hospitality industry lost their jobs during the lockdown in 2020, raising concerns about the mental health of its residents during the pandemic [9]. The pandemic struck mental health and substance use not only because of the disease itself, but also because of lockdowns, isolation, the economic downturn, and job losses [10]. The COVID-19 pandemic also reduced emergency department (ED) visits in the US, with the lowest-acuity ED visits having the most dramatic reduction, indicating that patients who did not require as much immediate medical attention were more likely to avoid going to EDs during the pandemic [11]; however, some conditions experienced proportionally smaller ED decreases, particularly those involving mental health and substance use [12,13]. The limited access to mental health care in Nevada [8] might push more patients with mental health-related conditions to EDs during crises, while the state also has limited ED resources compared with the national average [14]. Analyzing ED visits that have increased proportionally during the pandemic will prompt policies for caring for vulnerable patients during a public health emergency or crisis in order to prevent them from ending up in the ED, and Nevada statistics can provide insightful information on this matter.

The COVID-19 pandemic had a wave pattern. Depending on the wave and the state governors’ decisions, different policies were in place across the USA during the COVID-19 pandemic in 2020 and 2021 [15]. For example, in Nevada, COVID-19-related regulations were stricter in 2020 than in 2021 [16]. The COVID-19 pandemic literature on ED visits associated with mental health conditions and substance use tended to focus on the period early in the pandemic in 2020 [15,17,18,19,20,21,22,23]. Ridout and colleagues, in a study conducted early in the pandemic in 2020, found that a significant number of youth, especially women with no prior psychiatric history in North California, were admitted to the ED for suicide-associated issues [21]. A cross-sectional study of ED visits on children (5–17 years old) with a primary mental health diagnosis in the Chicago area found that visits for suicide or self-injury increased by 6.69% during the pandemic [22]. Venkatesh and colleagues, similar to Pines and colleagues, found that substance use-associated ED visits proportionally increased during the pandemic [15,17]. In a study inclusive of the Southern States, Petal and colleagues found an upsurge in opioid overdoses: 10.3% more opioid-associated deaths occurred from January to October 2020 than in 2019 [24]. Although these studies offer insightful information on the pandemic’s early stages and specific facilities, they have limited potential for generalization to other facilities and the rest of the pandemic period.

Mental health conditions and substance use are linked, and substance use can be regarded as a mental health condition that also affects behavior [25]. For example, the rates of cigarette smoking are approximately two- to four-fold higher in patients with a psychiatric disorder [5]. Some mental health conditions and substance use might be of particular concern due to their high prevalence and/or serious outcomes, as well as their reported increases during the pandemic [26,27,28]. Suicidal ideation, suicide attempts, and schizophrenia are all common and serious conditions that rose during the COVID-19 pandemic [27,28]. Uncontrolled schizophrenia has been associated with an increase in suicide attempts, and suicide is the tenth leading cause of death in the US [29,30]. Opioids, cannabis, alcohol, and cigarettes are commonly used substances in the US [27], and their use reportedly increased during the pandemic [12,31].

By determining the prevalence of certain conditions among ED patients during the pandemic, policymakers would be able to construct vital public health interventions intended to target a subset of the population with a higher possibility of ED visits during a crisis. This higher possibility could be due to the rising prevalence of these conditions in the general population or dwindling non-ED-facility options during lockdowns in the pandemic. Another explanation would be that these subsets, relative to the general population, might exhibit less behavioral caution in terms of ED visits during the pandemic. It is worth mentioning that between 13.7 and 27.1% of all ED visits in the USA could be unnecessary or treated at alternative sites [32]. Prior studies on mental health conditions and substance use associated with ED visits were conducted early in the pandemic, either in general (not on individual conditions) [20] or on just one condition [21]. The aim of this study was to compare potential changes in ED visits associated with common and/or serious mental health conditions (suicidal ideation, suicide attempts, and schizophrenia) and more commonly used substances (opioids, cannabis, alcohol, and cigarettes) during the COVID-19 pandemic in 2020 and 2021 as opposed to the pre-pandemic years. With this approach, this study attempted to examine whether the earlier effect of the pandemic would differ from the later effect on mental health and substance use among ED visits.

## 2. Methods

### 2.1. Data

The Nevada State Emergency Department Databases (SEDDN) containing all ED visits in 2018 and 2019 (two years before the pandemic), as well as 2020 and 2021 (two years since the pandemic), were used. The SEDDN contains rich information on all non-federal acute community hospitals in Nevada [26]. All ED visits associated with opioids, cannabis, cigarette smoking, alcohol use, suicidal ideation, suicide attempts, and schizophrenia were identified using the International Classification of Diseases, 10th Edition (ICD-10). These codes are listed in Appendix A and have been used in prior publications [26,33]. The University of Nevada, Las Vegas, institutional review board deemed this study exempt because the SEDDN database provides administrative data after complete de-identification [26]. For data analysis, a total of 4,185,416 ED visits (2018–2021) were included in this study. The demographics of the study population, as well as the seven variables’ frequencies from 2018 to 2021, are indicated in Table 1.

### 2.2. Measures and Data Analysis

Seven dichotomous dependent variables were studied here as follows: three common and serious mental health conditions, including suicidal ideation, suicide attempts, and schizophrenia, and four commonly used substances, including cigarette smoking, alcohol drinking, opioid use, and cannabis use. Age, gender, race/ethnicity, and payer source have been previously associated with these dependent variables [26] and were, therefore, included as independent variables in the regression model. In order to control for time and detect a potential trend, year was included as a dummy variable in all seven regression analyses, as used by other prior studies [34].

The patients’ age groups (<12, 12–17, 18–24, 25–34, 35–44 (reference), 45–54, 55–64, and ≥65), gender, payer source (Medicare, Medicaid, uninsured, other insurance, and private insurance (reference)), race/ethnicity (Black, Hispanic, Asian/Pacific Islander, White (reference), and others), and time (years 2018 (reference), 2019, 2020, and 2021) were the independent variables in each analysis [26].

Multiple visits from the same patient would be considered distinct ED visits because the data had been deidentified. As a result, the ED visits served as the unit of analysis [26]. To account for variations within hospitals due to the clustering effect, we utilized the generalized linear model for multivariable analysis and treated hospital as the random effect while estimating the fixed effect of the independent variables of individual hospital discharges [26]. All statistical analyses were conducted using SAS software version 9.4 (SAS Institute Inc.; Cary, NC, USA). *p*-values of <0.05 (2-tailed) were considered statistically significant.

## 3. Results

The numbers of ED visits were 1,107,950 (26.5%), 1,153,000 (27.5%), 924,887 (22.1), and 999,579 (23.2%) from 2018 to 2021, respectively, with a total number of 4,185,416 (Table 1). In all of these four years, more than 50% of ED visits were by women. Medicaid was the most prevalent payer source, covering more than 35% of ED visits. The proportion of White people who visited an ED decreased from 54.0% to 48.8%, whereas it increased for Black, Hispanic, and Asian people. Among all ED visits, the percentage of suicidal ideation was 1.69 in 2018, peaked at 1.96 in 2020, and decreased to 1.89 in 2023; the percentage of suicide attempts was 0.11 in 2018, peaked at 0.13 in 2020, and decreased to 0.12 in 2021; the percentage of schizophrenia was 1.09 in 2018, peaked at 1.87 in 2020, and decreased to 1.48 in 2021; the percentage of opioid use was 0.67 in 2018 and it peaked at 0.70 in 2020; the percentage of cannabis use was 1.26 in 2018 and peaked at 1.48 in 2020; the percentage of alcohol drinking was 3.33 in 2018 and peaked at 4.00 in 2020; and the percentage of smoking was 7.43 in 2018 and peaked at 9.67 in 2020 (Table 1). Generally, the rates of these conditions were higher in 2018 than in 2019 (Table 1). Therefore, 2018 was set as the reference year.

Table 2 indicates the factors associated with the mental health conditions of suicidal ideation, suicide attempts, and schizophrenia among ED visits in Nevada from 2018 to 2021. The odds of suicidal ideation-, suicide attempt-, and schizophrenia-associated ED visits were significantly higher during both years of the pandemic (2020 and 2021) compared with 2018. The adjusted odds of suicidal ideation-associated ED visits were 11% (95% CI = 1.04–1.19) and 9% (95% CI = 1.02–1.17) higher in 2020 and 2021, respectively, than those in 2018. The odds of suicide attempt-associated ED visits were 20% (95% CI = 1.09–1.33) and 16% (95% CI = 1.05–1.27) higher in 2020 and 2021, respectively, than those in 2018. The odds of schizophrenia-associated ED visits were 60% (95% CI = 1.47–1.75) and 28% (95% CI= 1.17–1.40) higher in 2020 and 2021, respectively, than those in 2018. The odds of suicidal ideation and schizophrenia were significantly 8% lower (95% CI = 0.86–0.99) and 23% higher in 2019 (95% CI = 1.12–1.34) than those in 2018, respectively. Other factors were also related to the odds of these mental health conditions for ED visits. ED visits associated with these three mental health conditions were significantly less likely to be female (suicidal ideation: OR = 0.43, 95% CI = 0.41–0.46; suicide attempts: OR = 0.927, 95% CI = 0.865–0.99; schizophrenia: OR = 0.33, 95% CI = 0.31–0.35). The age group of 12–17 years had significantly higher odds of suicidal ideation- (OR = 1.45, 95% CI = 1.32–1.59) and suicide attempt- (OR = 4.86, 95% CI = 4.32–5.46) associated ED visits compared with the control age group of 35–44 years. However, the control group had higher odds of schizophrenia-associated ED visits compared with the other five age groups (Table 2). Compared with the White race, the three other races (Black, Hispanic, and Asian) had significantly lower odds of suicidal ideation-, suicide attempt-, and schizophrenia-associated ED visits (Table 2), except for schizophrenia-associated ED visits for Black people, who had higher odds of schizophrenia-associated ED visits compared with White people (OR = 1.41, 95% CI = 1.32–1.52). Compared with private health insurance, both Medicaid and Medicare were significantly associated with higher odds of all three types of mental health-associated ED visits (Table 2).

Table 3 indicates the factors associated with the use of opioids, cannabis, alcohol, and cigarette smoking among ED visits in Nevada from 2018 to 2021. Opioid- and cannabis-associated ED visits had significantly lower odds in 2019 and 2021 compared with those in 2018. Cannabis-associated ED visits had significantly 11% higher odds in 2020 compared with 2018 (95% CI = 1.06–1.16). Cigarette smoking- and alcohol-drinking-associated ED visits had higher odds in 2020 and 2021 compared with 2018, and the highest odds were for smoking-associated ED visits in 2020 (OR = 1.27, 95% CI = 1.22–1.32). Women compared with men had lower odds of all these four conditions-associated ED visits (Table 3). Compared with the 35–44 age group, the 25–34 age group and the six other age groups had significantly higher and lower odds of opioid-associated ED visits, respectively. Cannabis-associated ED visits had higher odds in the 18–24- and 25–34-year age groups compared with the 35–44-year age group. None of the age groups had significantly higher odds of smoking-associated ED visits compared with the 35–44-year age group. Alcohol-drinking-associated ED visits had significantly higher odds in the age groups of 45–54 and 55–64 years compared with the age group of 35–44 years. Compared with the White race, the Black, Hispanic, and Asian races had significantly lower odds of opioid-, smoking-, and drinking-associated ED visits. Regarding cannabis-associated ED visits, only the Black race had significantly higher odds, while the other two races, i.e., Hispanic and Asian, had significantly lower odds compared with the White race. ED visits covered by Medicare and Medicaid, as well as those of the uninsured, had higher odds of opioid, cannabis, cigarette smoking, and alcohol use compared with ED visits covered by private insurance (Table 3).

## 4. Discussion

Here, we examined certain mental health and substance use conditions among ED visits in Nevada between 2018 and 2021 using multivariable analysis. We investigated 2020 and 2021 separately in order to comprehend how the various time points during the pandemic impacted ED visits associated with these conditions. We found that ED visits increased from 2018 to 2019, decreased in 2020, and increased again in 2021, but not to the pre-pandemic level. This trend is also consistent with the national trend in ED visits [35]. Generally, COVID-19-related lockdowns in Nevada were laxer in 2021 than they were in 2020 [16]. Despite the fact that there were more COVID-19 cases in 2021 than in 2020 [16], our findings suggest that the pandemic effects were possibly stronger in 2020 than those in 2021, but there are more details to our findings.

Most previous studies on mental health conditions-associated ED visits during the COVID-19 pandemic were limited to 2020 [20]. A study on one million non-COVID-19 ED visits in Missouri, USA, in 2020 found that the proportion of mental health conditions among all ED visits increased. They did not mention what mental health conditions were included in their study [20]. Using national data, Holland and colleagues found that suicide attempt-associated ED visits increased in 2020 compared with the pre-pandemic period, but they did not study those rates in 2021 [35]. We found that the odds of suicidal ideation-, suicide attempt-, and schizophrenia-associated ED visits significantly increased in 2020 and 2021 compared with 2018, with the highest odds in 2020, indicating a stronger earlier impact of the pandemic than that in 2021. Regarding suicide, Nevada is now ranked 12th in the US and is no longer among the top 10 states. With 642 fatalities and a rate of 19.8 suicides per 100,000 people, Nevada came in seventh place in 2019 [36]. Nevada’s Office of Suicide Prevention reports that, while suicide rates nationwide have increased, they have remained stable or even decreased in Nevada [36]. Neither suicidal ideation- nor suicide attempt-associated ED visits significantly increased in 2019 compared with 2018, indicating that their increase during the pandemic may not have been due to their possible gradual increase in Nevada. However, schizophrenia-associated ED visits significantly increased in 2019 compared with 2018. In our study, the odds ratio for schizophrenia in 2020 was the highest (Table 2). More research is needed to determine whether this highest odds ratio is related to the pandemic or its gradual increase. It has been reported that the frequency of schizophrenia among ED visits increased in the early pandemic, which might have been due to the increased need for emergency care for schizophrenia patients [13] or an increase in the incidence of the disease.

The COVID-19 pandemic has been associated with increased alcohol consumption and cigarette smoking [10]. We found that the odds of alcohol and cigarette smoking use among ED visits significantly increased in 2020, and 2021 compared with 2018, with the highest odds in 2020, another indication of the strong early impact of the pandemic. Consistently, data on ED visits in the Washington, DC/Baltimore, and Maryland areas in 2019 and 2020 revealed that a higher percentage of patients reported alcohol drinking during the pandemic [12]. Another study in Ontario, Canada, found that the proportion of all-cause ED visits due to alcohol increased by 11.4% [37]. A study from Minnesota, USA, indicated a significant increase in ED visits among smokers [38]. Alcohol use and cigarette smoking have been consistently reported to have increased among ED visits during the COVID-19 pandemic [37,38], which could be a result of their increased rates during the pandemic [10] and/or possible increased emergency conditions among its users.

Opioids and cannabis had been subject to new rules and legislation at the federal and state levels prior to the pandemic [26]. Beginning in 2010, there was a decrease in opioid-associated ED visits, which coincided with federal initiatives calling for more prudent opioid prescription [26]. Policies regarding cannabis consumption mainly depend on the State authorities [26]. Nevada legalized cannabis for both medical and recreational use in 2001 and 2016 [26,39,40]. The legal use of medical and recreational cannabis went into effect in 2013 and 2017, respectively, while stricter opioid prescription laws went into effect in Nevada in 2018 [39,40]. We found that opioid-associated ED visits did not significantly increase during the pandemic compared with those in 2018, and even significantly decreased in 2021 and 2019 compared with 2018, which might be related to the strict opioid prescription laws in Nevada [39,40]. It was supposed to be the case that opioid use in all three years of 2019, 2020, and 2021 should have been lower than that in 2018. The lack of a difference between 2020 and 2018, however, may indicate that the strong early impact of the pandemic offset the strict opioid prescription laws starting in 2018 [39,40]. Using national data, Holland and colleagues found that opioid overdose-associated ED visits increased in 2020 compared with the pre-pandemic period, but they did not study those rates in 2021 [35]. Patel and colleagues investigated patients presenting to the ED with opioid overdoses at the University of Alabama in 2019 and 2020. They reported an increase in opioid overdose visits in 2020 compared with 2019 [24]. The differences between our results and those of others might be due to the use of different inclusion criteria. Our inclusion criteria were opioid use, but in other studies, their inclusion criteria were opioid overdoses [24,35]. Deaths due to opioid overdoses decreased in Nevada from 2018 to 2019, but increased from 2019 to 2020 according to a report by the Centers for Disease Control and Prevention [41].

Less information is available regarding cannabis-associated ED visits. The proportion of cannabis-associated ED visits was previously found to increase among children in 2019 compared with 2020 [2]. We found that cannabis-associated ED visits significantly increased in 2020 and decreased in 2021 compared with 2018. Nevada statistics indicate that cannabis sales increased from 2018 to 2021 [42]. Therefore, the decrease in cannabis-associated ED visits may not be due to lower consumption. More behavioral research is needed to investigate the underlying causes of this decrease and whether it will persist in the future. According to recent data, recreational cannabis legalization may be a harm-reduction strategy to combat the opioid epidemic and has been associated with reduced opioid-related ED visits, particularly among men and adults between the ages of 25 and 44 [43]. Future studies will determine whether the lack of a significant increase in opioid-related ED visits in Nevada can be attributed to the harm-reduction effects of recreational cannabis legalization.

Our study has some limitations. We identified the seven mental health and substance-use conditions using ICD codes. Any coding error could have resulted in the misclassification of ED visits. There are also other common and serious mental health and substance-use conditions. Notably, uncontrolled depression, anxiety disorders, bipolar disorders, and other mental health conditions can all contribute to suicide and have been exacerbated by the COVID-19 pandemic [12,13,17,18,19]. We were not able to accommodate all of these important conditions in our study, which aimed to investigate each condition separately, rather than combining them. Furthermore, we were unable to account for all potential confounding factors. For example, opioid use has been subject to strict federal and state regulations [26], but we did not account for this in our regression model. We analyzed the odds of certain conditions over the year. The increase in odds indicates that the proportion of ED visits for one condition versus others is higher in a given year, but it does not necessarily imply that the number of that specific type of ED visit has increased over the year. In our analysis, however, the observed increases in odds were often accompanied by corresponding increases in numbers (Table 1).

Our findings add to the literature as we analyzed longer periods during the COVID-19 pandemic and covered mental health- and substance use-associated ED visits in 2020 and 2021, whereas previous studies mainly focused on 2020 [35], from which both the early impact and the later impact of the pandemic were examined. Our findings indicate a stronger early impact than a later one. We also specifically investigated serious mental health conditions, rather than less serious mood disorders [20]. Further, we specifically looked at different mental health and substance use conditions separately, which is very important for making informed decisions. Policymakers need to know what mental health conditions or substances need more attention.

## 5. Conclusions

In conclusion, among the seven common mental health conditions and substance use-associated ED visits in Nevada, suicidal ideation, suicide attempts, schizophrenia, cigarette smoking, and alcohol drinking had significantly higher odds in both 2020 and 2021 compared with the pre-pandemic period. Cannabis-associated ED visits had significantly higher odds only in 2020 compared with the pre-pandemic period, but opioid-associated ED visits did not have significantly higher odds during the pandemic compared with the pre-pandemic period. It was observed that the pandemic had stronger early effects on mental health and the use of substances, and the effects may have decreased as the pandemic continued. This information can help policymakers to better comprehend how the outbreak affected society both stateside and nationally, make informed decisions, and develop effective policy measures and programs to prepare an effective response to large-scale public health emergencies, such as the COVID-19 pandemic, at their early stages.

## Figures and Tables

**Table 1 ijerph-20-04389-t001:** Patient characteristics, mental health conditions, and use of substances of emergency department (ED) visits in Nevada (2018–2021).

Characteristic	2018	2019	2020	2021
Number of ED visits, n	1,107,950	1,153,000	924,887	999,579
Age, mean (St.d.) ^⁋^, years	38.4 (23.0)	38.5 (23.1)	40.4 (22.0)	39.5 (22.3)
Gender n (%) *				
Women	606,721 (54.8)	629,789 (54.6)	495,011 (53.5)	540,215 (54.0)
Men	501,121 (45.2)	523,105 (45.4)	429,793 (46.5)	459,272 (45.9)
Race/ethnicity n (%) *				
White	598,386 (54.1)	598,654 (51.9)	478,805 (51.8)	487,453 (48.8)
Hispanic	144,801 (13.1)	158,373 (13.7)	121,676 (13.2)	140,521 (14.1)
Black	209,026 (18.9)	222,514 (19.3)	181,597 (19.6)	211,872 (21.2)
Asians	41,133 (3.7)	43,589 (3.8)	35,043 (3.8)	42,163 (4.2)
Others	114,604 (10.3)	129,870 (11.3)	107,766 (11.6)	117,570 (11.7)
Payer source n (%) *				
Private insurance	353,464 (31.9)	365,490 (31.7)	298,678 (32.3)	298,678 (32.8)
Medicaid	416,205 (37.6)	430,550 (37.3)	337,225 (36.5)	380,201 (38.0)
Medicare	193,348 (17.5)	197,839 (17.2)	160,015 (17.3)	160,110 (16.0)
Self-pay	105,498 (9.5)	116,877 (10.1)	91,280 (9.9)	90,605 (9.1)
No charge	23,471 (2.1)	18,639 (1.6)	13,849 (1.5)	9306 (0.9)
Others	15,964 (1.4)	23,605 (2.1)	23,840 (2.6)	31,323 (3.1)
Mental health conditions n (%) *				
Suicidal ideation-associated ED visits	18,671 (1.69)	17,786 (1.54)	18,092 (1.96)	18,905 (1.89)
Suicide attempt-associated ED visits	1118 (0.11)	1117 (0.10)	1181 (0.13)	1223 (0.12)
Schizophrenia-associated ED visits	12,115 (1.09)	15,190 (1.32)	17,280 (1.87)	14,761 (1.48)
Use of substances n (%) *				
Opioid-associated ED visits	7475 (0.67)	7025 (0.61)	6519 (0.70)	6265 (0.63)
Cannabis-associated ED visits	13,966 (1.26)	13,759 (1.19)	13,690 (1.48)	11,750 (1.17)
Alcohol-drinking-associated ED visits	36,930 (3.33)	38,976 (3.38)	37,031 (4.00)	37,760 (3.78)
Smoking-associated ED visits	82,271 (7.43)	101,301 (8.79)	89,402 (9.67)	79,074 (7.91)

^⁋^ St.d.; standard deviation; * % among all ED visits in that particular year.

**Table 2 ijerph-20-04389-t002:** Results of multivariable regression analyses on suicidal ideation, suicide attempts, and schizophrenia among emergency department (ED) visits in Nevada (2018–2021).

	Suicidal Ideation			Suicide Attempt			Schizophrenia		
Independent Variable	OR ⁋	95% CI ⁌	*p*-Value	OR ⁋	95% CI ⁌	*p*-Value	OR ⁋	95% CI ⁌	*p*-Value
2018 (ref) *									
2019	0.92	0.86–0.99	0.0289	0.92	0.83–1.01	0.0883	1.23	1.12–1.34	<0.0001
2020	1.11	1.04–1.19	0.0012	1.20	1.09–1.33	0.0001	1.60	1.47–1.75	<0.0001
2021	1.09	1.02–1.17	0.0057	1.16	1.05–1.27	0.0024	1.28	1.17–1.40	<0.0001
2018 (ref) *									
Age groups									
35–44 (Ref) *									
<12	0.05	0.04–0.07	<0.0001	0.16	0.12–0.20	<0.0001	0.003	<0.001–0.008	<0.0001
12–17	1.45	1.32–1.59	<0.0001	4.86	4.32–5.46	<0.0001	0.08	0.06–0.12	<0.0001
18–24	0.95	0.87–1.04	0.2981	2.46	2.19–2.76	<0.0001	0.54	0.48–0.61	<0.0001
25–34	0.957	0.89–1.02	0.2249	1.31	1.17–1.47	<0.0001	0.95	0.87–1.03	0.2639
45–54	0.802	0.74–0.86	<0.0001	0.56	0.48–0.66	<0.0001	0.79	0.72–0.86	<0.0001
55–64	0.513	0.46–0.56	<0.0001	0.28	0.23–0.34	<0.0001	0.47	0.43–0.53	<0.0001
≥65	0.093	0.08–0.10	<0.0001	0.10	0.08–0.14	<0.0001	0.07	0.05–0.08	<0.0001
Gender									
Men (ref) *									
Women	0.43	0.41–0.46	<0.0001	0.92	0.86–0.99	0.0330	0.33	0.31–0.35	<0.0001
Race/ethnicity									
White (ref) *									
Black	0.83	0.78–0.89	<0.0001	0.41	0.36–0.45	<0.0001	1.41	1.32–1.52	<0.0001
Hispanic	0.63	0.58–0.69	<0.0001	0.86	0.78–0.95	0.0028	0.86	0.77–0.96	0.0076
Asian	0.68	0.58–0.80	<0.0001	0.80	0.65–0.97	0.0287	0.67	0.53–0.84	0.0008
Payer source									
Private insurance (Ref) *									
Medicare	3.25	2.95–3.57	<0.0001	1.50	1.25–1.80	<0.0001	7.42	6.59–8.37	<0.0001
Medicaid	2.29	2.15–2.44	<0.0001	1.14	1.05–1.24	0.0009	4.19	3.80–4.61	<0.0001
Uninsured	1.02	0.92–1.12	0.6348	1.09	0.98–1.22	0.1092	1.11	0.96–1.29	0.1449

⁋ OR, Odds ratio; ⁌ CI, confidence interval; * ref, reference group.

**Table 3 ijerph-20-04389-t003:** Results of multivariable regression analyses on opioid, cannabis, cigarette, and alcohol use among emergency department (ED) visits in Nevada (2018–2021).

	Opioids			Cannabis			Cigarette Smoking			Alcohol		
Independent Variable	OR ⁋	95% CI ⁌	*p*-Value	OR ⁋	95% CI ⁌	*p*-Value	OR ⁋	95% CI ⁌	*p*-Value	OR ⁋	95% CI ⁌	*p*-Value
Years												
2018 (ref) *												
2019	0.92	0.86–0.98	0.0122	0.95	0.91–0.99	0.0428	1.23	1.19–1.28	<0.0001	1.02	0.96–1.08	0.4408
2020	1.00	0.93–1.06	0.9979	1.11	1.06–1.16	<0.0001	1.27	1.22–1.32	<0.0001	1.12	1.05–1.19	0.0002
2021	0.93	0.87–0.99	0.0405	0.89	0.85–0.93	<0.0001	1.05	1.01–1.09	0.0056	1.11	1.04–1.18	0.0005
Age groups												
35–44 (Ref) *												
<12	0.01	0.00–0.02	<0.0001	0.02	0.01–0.02	<0.0001	0.003	0.002–0.005	<0.0001	0.002	0.001–0.006	<0.0001
12–17	0.07	0.05–0.10	<0.0001	0.75	0.69–0.82	<0.0001	0.04	0.03–0.05	<0.0001	0.14	0.11–0.17	<0.0001
18–24	0.62	0.57–0.68	<0.0001	1.56	1.48–1.63	<0.0001	0.45	0.42–0.47	<0.0001	0.59	0.54–0.64	<0.0001
25–34	1.14	1.07–1.22	<0.0001	1.30	1.24–1.36	<0.0001	0.79	0.76–0.83	<0.0001	0.80	0.75–0.86	<0.0001
45–54	0.75	0.70–0.82	<0.0001	0.67	0.64–0.71	<0.0001	0.99	0.95–1.03	0.8644	1.08	1.01–1.15	0.0219
55–64	0.78	0.72–0.85	<0.0001	0.50	0.47–0.54	<0.0001	0.94	0.90–0.98	0.0065	1.31	1.22–1.40	<0.0001
≥65	0.32	0.29–0.36	<0.0001	0.16	0.15–0.18	<0.0001	0.33	0.31–0.35	<0.0001	0.46	0.42–0.51	<0.0001
Gender												
Men (ref) *												
Women	0.68	0.65–0.71	<0.0001	0.57	0.55–0.59	<0.0001	0.73	0.71–0.75	<0.0001	0.36	0.35–0.38	<0.0001
Race/ethnicity												
White (ref) *												
Black	0.45	0.42–0.48	<0.0001	1.29	1.25–1.34	<0.0001	0.74	0.72–0.77	<0.0001	0.53	0.50–0.57	<0.0001
Hispanic	0.46	0.42–0.50	<0.0001	0.67	0.63–0.71	<0.0001	0.38	0.36–0.40	<0.0001	0.77	0.72–0.83	<0.0001
Asian	0.30	0.24–0.36	<0.0001	0.57	0.51–0.64	<0.0001	0.38	0.34–0.41	<0.0001	0.48	0.41–0.56	<0.0001
Payer source												
Private insurance (Ref) *												
Medicare	2.82	2.58–3.09	<0.0001	1.31	1.21–1.40	<0.0001	1.67	1.59–1.75	<0.0001	1.17	1.07–1.28	0.0005
Medicaid	2.61	2.45–2.79	<0.0001	1.29	1.24–1.34	<0.0001	2.07	2.00–2.14	<0.0001	2.16	2.04–2.29	<0.0001
Uninsured	1.67	1.53–1.82	<0.0001	1.37	1.30–1.44	<0.0001	1.30	1.24–1.36	<0.0001	2.67	2.49–2.85	<0.0001

⁋ OR, Odds ratio; ⁌ CI, confidence interval; * ref, reference group.

## Data Availability

Data was obtained from the Center for Health Information Analysis for Nevada and are available from Jay J. Shen with the permission of Nevada Department of Health and Human Services.

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
