# Peer review of "Mental Health Conditions– and Substance Use—Associated Emergency Department Visits during the COVID-19 Pandemic in Nevada, USA"

_ijerph, 2023, doi:10.3390/ijerph20054389_

Round 1

Reviewer 1 Report

Thank you for the opportunity to review your manuscript titled sever mental health statuses and substance use associated emergency department visits during the covid-19 pandemic in Nevada, USA. Below are some suggestions to improve the quality and clarity of your manuscript.

Title:

1.       Consider changing status to diagnoses or conditions in the title for consistency with the manuscript.

Introduction:

2.       Consider explaining why emergency department visits were down in Nevada during the pandemic.

3.       In the sentence of Nevada’s rank for mental illness please add clarity to the sentence to explain Nevada ranked 44/51 due to high prevalence of mental illness.

4.       Consider deleting the sentence describing no sources are available describing the prevalence of mental illness in Nevada during the Covid-19 pandemic. Consider adding national statistics that support the premise the Covid-19 pandemic worsened mental illness overall.

5.       It is unclear to me why schizophrenia is included with suicide and suicidal ideation. While a serious condition that requires a lifetime of medical management and was likely worsened by the pandemic does not necessarily make sense to include with suicide. Uncontrolled depression, anxiety disorders, substance use disorders, schizoaffective disorders, bipolar disorders, ADHD, and other mental health conditions can contribute to suicide and were worsened by the COVID-19 pandemic. Consider revising the Introduction to explain why schizophrenia should be highlighted in this study.

Methods:

1.      Consider redefining the conditions as severe mental health conditions. Authors should consider in future investigations “serious mental illness” and the mental health diagnoses that are included in this definition.

2.       Why were the independent variables selected in your Methods?

3.       What software was used for analyses and what p values were used to determine significance?

4.       Is more than one diagnosis code associated with each individual ED visit? Or is the only diagnosis code that of the primary diagnosis?

5.       It may be inappropriate to include the year of the ED visit within the regressions. The proportion of ED visits within a year is irrelevant to know given other ED visits within the year are unconnected and unrelated events subject to a number of confounders authors are not controlling for. ED visits are typically assessed as count data and analyzed as such when evaluating longitudinal data. The change in the rate of targeted ED visits (substance abuse and suicides) year over year is more interesting and more relevant for authors to know. I recommend authors exclude year from the logistic regression to focus on what characteristics of patients stand out as predictors of the events and independently assess change in the rate of targeted ED visits from one year to the next.

6.       Authors appear to have text within this manuscript that is identical to other text authors are affiliated with. Please cite the work as this may be flagged as plagiarism. (https://www.ncbi.nlm.nih.gov/pmc/articles/PMC6882558/)

Limitations to note:

1.       Authors do not discuss limitations to their analysis or approach. Please add a limitations section. I am unable to see the supplemental information, please make sure the diagnosis codes are derived from citable sources. Please describe if diagnosis codes utilized have been validated or not. Please discuss confounders that can impact your results.

Author Response

Dear Editor,

Dear Reviewers,

We appreciate your insightful comments. Please find a point-by-point response to these comments below:

Reviewer 1:

Title:

  1. Consider changing status to diagnoses or conditions in the title for consistency with the manuscript.

Reply: It was changed to "conditions" and highlighted.

Introduction:

  1. Consider explaining why emergency department visits were down in Nevada during the pandemic.

Reply: The following sentences were added to the introduction:

To the first paragraph of the introduction section:

The COVID-19 pandemic also reduced emergency department (ED) visits in the US [10], with the lowest acuity ED visits having the most dramatic reduction, indicating that patients who did not require as much immediate medical attention were more likely to avoid going to EDs during the pandemic [11]

To the fourth paragraph of the introduction section:

It is worth mentioning that between 13.7 and 27.1% of all ED visits in the USA could be unnecessary or treated at alternative sites [32].

  1. In the sentence of Nevada’s rank for mental illness please add clarity to the sentence to explain Nevada ranked 44/51 due to high prevalence of mental illness.

Reply: The following was revised as below:

Nevada ranks 44th out of 51 states (of the USA) in terms of the prevalence of mental illnesses [7]

  1. Consider deleting the sentence describing no sources are available describing the prevalence of mental illness in Nevada during the Covid-19 pandemic. Consider adding national statistics that support the premise the Covid-19 pandemic worsened mental illness overall.

Reply: Thank you for the comment. The sentence was deleted, and the first paragraph of the introduction was reordered and revised as below:

Mental health and substance use issues are intertwined, and both reportedly increased in the United States and globally during the COVID-19 pandemic [1-5]. Nevada has distinct characteristics in terms of mental health and substance use, with rates that are typically higher than the national average [6]. Nevada ranks 44th out of 51 states (of the USA) in terms of the prevalence of mental illnesses [7]. Nevada is also one of the worst states for access to mental health care services (39 among 51 states) [8]. The pandemic had a significant economic impact on Nevada, a world tourism center. More than 90% of those employed in the hospitality industry lost their jobs during the lockdown in 2020, raising concerns about the mental health of its residents during the pandemic [9]. The pandemic struck mental health and substance use not only because of the disease itself but also because of lockdowns, isolation, the economic downturn, and job losses [1-4]. The COVID-19 pandemic also reduced emergency department (ED) visits in the US [10], with the lowest acuity ED visits having the most dramatic reduction, indicating that patients who did not require as much immediate medical attention were more likely to avoid going to EDs during the pandemic [11]; however, some conditions experienced proportionally smaller ED decreases, particularly those involving mental health and substance use [12, 13]. Limited access to mental health care in Nevada [8] might push more patients with mental health related conditions to EDs during crises, while the state also has limited ED resources compared to the national average [14]. Analyzing ED visits that have increased proportionally during the pandemic will prompt policies for caring for vulnerable patients during a public health emergency or crisis in order to prevent them from ending up in the ED, and Nevada statistics can provide insightful information on this matter.

  1. It is unclear to me why schizophrenia is included with suicide and suicidal ideation. While a serious condition that requires a lifetime of medical management and was likely worsened by the pandemic does not necessarily make sense to include with suicide. Uncontrolled depression, anxiety disorders, substance use disorders, schizoaffective disorders, bipolar disorders, ADHD, and other mental health conditions can contribute to suicide and were worsened by the COVID-19 pandemic. Consider revising the Introduction to explain why schizophrenia should be highlighted in this study.

Reply: Thank you for the comment. Some sentences were added to the limitation of the study and some sentences were revised in the third paragraph of the introduction section:

The third paragraph of the introduction section:

Mental health conditions and substance use are linked, and substance use can be regarded as a mental health condition that affects behavior as well [25]. For example, rates of cigarette smoking are approximately two- to four-fold higher in patients with a psychiatric disorder [5]. Some mental health conditions and substance use might be of particular concern due to their high prevalence and/or serious outcomes, as well as reported increases during the pandemic [26-28]. Suicidal ideation, suicidal attempts, and schizophrenia are all common and serious conditions that have risen in the COVID-19 pandemic [27, 28]. Uncontrolled schizophrenia has been associated with an increase in suicidal attempts, and suicide is the tenth leading cause of death in the US [29, 30]. Opioids, cannabis, alcohol, and cigarette smoking are commonly used substances in the US [27], and their use reportedly increased during the pandemic [12, 31].  

The limitation in the discussion section:

There are also other common and serious mental health and substance use conditions. Notably, uncontrolled depression, anxiety disorders, bipolar disorders, and other mental health conditions can all contribute to suicide and were exacerbated by the COVID-19 pandemic [12, 13, 17-19]. We were not able to accommodate all these important conditions in our study, which aimed to investigate each condition separately rather than combining them.

Methods:

  1. Consider redefining the conditions as severe mental health conditions. Authors should consider in future investigations “serious mental illness” and the mental health diagnoses that are included in this definition.

Reply: Our definitions from mental health illnesses are based on the ICD codes, which have been used in prior studies. The following sentence and reference were added to the method section (2.1. data):

These codes were listed in Supplemental Table1 and have been used in prior publications [26, 33].

  1. Why were the independent variables selected in your Methods?

Reply: The following sentence was added to the method section (2.2. Measures and data analysis):

Age, gender, race/ethnicity, and payer source have been previously associated with these dependent variables [26] and therefore were included as independent variables in the regression model. In order to control for time and detect a potential trend, year was included as a dummy variable in all seven regression analyses, as used by other prior studies [34].

  1. What software was used for analyses and what p values were used to determine significance?

Reply: The following sentence was added to the method section (2.2. Measures and data analysis):

All statistical analysis were conducted using the SAS software version 9.4 (SAS Institute Inc., Cary, NC, USA). P-values < 0.05 (2-tailed) were considered statistically significant.

  1. Is more than one diagnosis code associated with each individual ED visit? Or is the only diagnosis code that of the primary diagnosis?

Reply: The goal of our study was to investigate the prevalence of the seven mental health and substance use dependent variables among ED visits. Our ICD-codes are associated with ED visits, and they are not necessary the primary diagnosis.

  1. It may be inappropriate to include the year of the ED visit within the regressions. The proportion of ED visits within a year is irrelevant to know given other ED visits within the year are unconnected and unrelated events subject to a number of confounders authors are not controlling for. ED visits are typically assessed as count data and analyzed as such when evaluating longitudinal data. The change in the rate of targeted ED visits (substance abuse and suicides) year over year is more interesting and more relevant for authors to know. I recommend authors exclude year from the logistic regression to focus on what characteristics of patients stand out as predictors of the events and independently assess change in the rate of targeted ED visits from one year to the next.

Reply: We revised Table 1 and included the frequencies of all conditions and variables (please see Table 1 in the manuscript). Since the goal of our study was examine a potential change in the trend of ED visits over years due to the COVID-19, year was included in the regression analysis, which has been used by other researchers. We added the following sentence and reference to the method section (2.2. Measures and data analysis).

In order to control for time and detect a potential trend, year was included as a dummy variable in all seven regression analyses, as used by other prior studies [34].

Sun Z, Laporte A, Guerriere DN, Coyte PC: Utilisation of home-based physician, nurse and personal support worker services within a palliative care programme in Ontario, Canada: trends over 2005-2015. Health Soc Care Community 2017, 25(3):1127-1138.

  1. Authors appear to have text within this manuscript that is identical to other text authors are affiliated with. Please cite the work as this may be flagged as plagiarism. (https://www.ncbi.nlm.nih.gov/pmc/articles/PMC6882558/)

Reply: The text had citations at the end of paragraph. The whole paragraph was revised as the below, and citation was added to the middle of paragraph as well.

Multiple visits from the same patient would be considered distinct ED visits because the data had been deidentified. As a result, the ED visits served as the unit of analysis [26]. To account for variations within hospitals due to the clustering effect, we utilized the generalized linear model for multivariable analysis and treated hospital as the random effect while estimating the fixed effect of the independent variables of individual hospital discharges [26].

Limitations to note:

  1. Authors do not discuss limitations to their analysis or approach. Please add a limitations section. I am unable to see the supplemental information, please make sure the diagnosis codes are derived from citable sources. Please describe if diagnosis codes utilized have been validated or not. Please discuss confounders that can impact your results.

Reply: Limitation was added. Supplementary file was provided, and ICD codes were cited.

The following sentence and reference were added to the method section (2.1. data):

These codes were listed in Supplemental Table1 and have been used in prior publications [26, 33].

Limitation:

Our study has some limitations. We identified the seven mental health and substance use conditions using ICD codes. Any coding error can result in the misclassification of ED visits. There are also other common and serious mental health and substance use conditions. Notably, uncontrolled depression, anxiety disorders, bipolar disorders, and other mental health conditions can all contribute to suicide and were exacerbated by the COVID-19 pandemic [12, 13, 17-19]. We were not able to accommodate all these important conditions in our study, which aimed to investigate each condition separately rather than combining them. Furthermore, we were unable to account for all potential confounding factors. For example, opioid use has been subject to strict federal and state regulations [26], but we did not account for this in our regression model.

Reviewer 2 Report

Mojtahedi and colleagues present work investigating the influence of the COVID-19 pandemic on emergency department presentations in the US state of Nevada. The authors investigate the association of mental health diagnosis and substance use (separately) with ED presentation. Results indicated increased ED attendance associated with suicidal ideation, attempt or schizophrenia. The associations with substance use were more mixed.

I have a number of concerns that prevent me from being able to endorse publication at this point.

Firstly, the impetus for the study is absent or missing. It is not clear why the authors examined the specific variables that they have. Why is there an expectation that ED presentations associated with serious mental illness would increase during the COVID-19 pandemic?

Similarly, the choice of variables seems random in nature. The authors appear to pluck suicidal ideation, suicidal attempts and schizophrenia as their "severe mental health conditions" without any justification. Can suicidal ideation be considered a mental health "condition"? Why schizophrenia and no other psychotic disorders? Is major depressive disorder any less severe? What of the personality disorders? The choice of mental health conditions is unmotivated.

The choice of substances also appears to run counter to the way that mental health conditions were selected. For the latter they chosen due to some perceived severity. The former were, instead, chosen due to their common presentation. These decisions seem at odds with one another.

The authors conducted 7 logistic regression models but it is not clear if a correction for familywise error was applied. Can the authors please clarify.

There was a sharp decline in ED attendances when the COVID-19 pandemic began. Would this large shift in baseline not potentially explain the increased association with the severe mental illnesses that were chosen for analysis? Schizophrenia and suicide seem more difficult to delay visiting an ED for than other ICD codes might be.

The results for the substances change sign or significance from year to year, appearing somewhat spurious in nature. The authors do not interpret this sufficiently, failing to provide a plausible cause for these patterns. Why, for example, are OR significantly increased for cannabis in 2020 but then decreased in 2021?

Author Response

Dear Editor,

Dear Reviewers,

We appreciate your insightful comments. Please find a point-by-point response to these comments below:

Reviewer 2

-Firstly, the impetus for the study is absent or missing. It is not clear why the authors examined the specific variables that they have. Why is there an expectation that ED presentations associated with serious mental illness would increase during the COVID-19 pandemic?

Reply:

Thank you for the comment. The first paragraph of the introduction section was revised, and some sentences were added to the last paragraph of the introduction section:

 The first paragraph of introduction section:

Mental health and substance use issues are intertwined, and both reportedly increased in the United States and globally during the COVID-19 pandemic [1-5]. Nevada has distinct characteristics in terms of mental health and substance use, with rates that are typically higher than the national average [6]. Nevada ranks 44th out of 51 states (of the USA) in terms of the prevalence of mental illnesses [7]. Nevada is also one of the worst states for access to mental health care services (39 among 51 states) [8]. The pandemic had a significant economic impact on Nevada, a world tourism center. More than 90% of those employed in the hospitality industry lost their jobs during the lockdown in 2020, raising concerns about the mental health of its residents during the pandemic [9]. The pandemic struck mental health and substance use not only because of the disease itself but also because of lockdowns, isolation, the economic downturn, and job losses [1-4]. The COVID-19 pandemic also reduced emergency department (ED) visits in the US [10], with the lowest acuity ED visits having the most dramatic reduction, indicating that patients who did not require as much immediate medical attention were more likely to avoid going to EDs during the pandemic [11]; however, some conditions experienced proportionally smaller ED decreases, particularly those involving mental health and substance use [12, 13]. Limited access to mental health care in Nevada [8] might push more patients with mental health related conditions to EDs during crises, while the state also has limited ED resources compared to the national average [14]. Analyzing ED visits that have increased proportionally during the pandemic will prompt policies for caring for vulnerable patients during a public health emergency or crisis in order to prevent them from ending up in the ED, and Nevada statistics can provide insightful information on this matter.

To the last paragraph of the introduction section:

By determining the prevalence of certain conditions among ED patients during the pandemic, policymakers would be able to construct vital public health interventions that are intended to target a subset of the population with a higher possibility of ED visits during a crisis. This higher possibility could be due to the rising prevalence of these conditions in the general population or dwindling non-ED facility options during lockdown in the pandemic. Another explanation would be that these subsets, relative to the general population, might exhibit less behavioral caution in terms of ED visits during the pandemic. It is worth mentioning that between 13.7 and 27.1% of all ED visits in the USA could be unnecessary or treated at alternative sites [32]. Prior studies on mental health conditions and substance use associated with ED visits were conducted early in the pandemic, either in general (not on individual conditions) [20] or on just one condition [21].

-Similarly, the choice of variables seems random in nature. The authors appear to pluck suicidal ideation, suicidal attempts and schizophrenia as their "severe mental health conditions" without any justification. Can suicidal ideation be considered a mental health "condition"? Why schizophrenia and no other psychotic disorders? Is major depressive disorder any less severe? What of the personality disorders? The choice of mental health conditions is unmotivated.

Reply: The following sentences were revised in the third paragraph of the introduction. and some sentences were added to the limitation section:

The third paragraph of the introduction:

Mental health conditions and substance use are linked, and substance use can be regarded as a mental health condition that affects behavior as well [25]. For example, rates of cigarette smoking are approximately two- to four-fold higher in patients with a psychiatric disorder [5]. Some mental health conditions and substance use might be of particular concern due to their high prevalence and/or serious outcomes, as well as reported increases during the pandemic [26-28]. Suicidal ideation, suicidal attempts, and schizophrenia are all common and serious conditions that have risen in the COVID-19 pandemic [27, 28]. Uncontrolled schizophrenia has been associated with an increase in suicidal attempts, and suicide is the tenth leading cause of death in the US [29, 30]. Opioids, cannabis, alcohol, and cigarette smoking are commonly used substances in the US [27], and their use reportedly increased during the pandemic [12, 31].  

To the limitation:

There are also other common and serious mental health and substance use conditions. Notably, uncontrolled depression, anxiety disorders, bipolar disorders, and other mental health conditions can all contribute to suicide and were exacerbated by the COVID-19 pandemic [12, 13, 17-19]. We were not able to accommodate all these important conditions in our study, which aimed to investigate each condition separately rather than combining them.

-The choice of substances also appears to run counter to the way that mental health conditions were selected. For the latter they chosen due to some perceived severity. The former were, instead, chosen due to their common presentation. These decisions seem at odds with one another.

Reply: We appreciate the comment. Since the mental health conditions studies here are also common, more emphasis was placed on variables being common as explained to the prior comment.

-The authors conducted 7 logistic regression models but it is not clear if a correction for familywise error was applied. Can the authors please clarify.

Reply:  The seven dependent variables were not directly related. Therefore, we did not perform familywise correction. If they were related like tumor stages, adding those corrections would be necessary.

-There was a sharp decline in ED attendances when the COVID-19 pandemic began. Would this large shift in baseline not potentially explain the increased association with the severe mental illnesses that were chosen for analysis? Schizophrenia and suicide seem more difficult to delay visiting an ED for than other ICD codes might be.

Reply: Thank you for the comment.  One sentence in the second paragraph of  discussion section was highlighted, and the following sentence was added to the last paragraph of

Introduction section:

The last paragraph of Introduction section:

This higher possibility could be due to the rising prevalence of these conditions in the general population or dwindling non-ED facility options during lockdown in the pandemic

The following sentence was highlighted in the discussion section:

It has been reported that the frequency of schizophrenia among ED visits has increased in the early pandemic that might be due to the increased need for emergency care for schizophrenia patients [13]

-The results for the substances change sign or significance from year to year, appearing somewhat spurious in nature. The authors do not interpret this sufficiently, failing to provide a plausible cause for these patterns. Why, for example, are OR significantly increased for cannabis in 2020 but then decreased in 2021?

Reply: Some sentences were moved from the introduction to the fourth paragraph of the discussion section, and some sentences were added to the fifth paragraph of the discussion section:

Fourth paragraph of the discussion section:

Opioid and cannabis had been subject to new rules and legislation at the federal and state levels prior to the pandemic [26]. Beginning in 2010, there was a decrease in opioid-associated ED visits, which coincided with federal initiatives calling for more prudent opioid prescribing [26]. Policy regarding cannabis consumption mainly depends on the State authorities [26]. Nevada legalized cannabis for both medical and recreational use in 2001 and 2016 [26, 40, 41]. Legal use of medical and recreational cannabis went into effect in 2013 and 2017, respectively, while stricter opioid prescription laws went into effect in Nevada in 2018 [40, 41].

 The fifth paragraph of the discussion section:

Nevada statistics indicate that cannabis sale increased from 2018 to 2021 [44]. Therefore, the decrease in cannabis-related ED visits may not be due to its lower consumption. More behavioral research is needed to investigate the underlying causes of this decrease and whether it will persist in the future.

Round 2

Reviewer 1 Report

Thank you for addressing my concerns in the revision. I appreciate the work the reviewers have done. I still do not agree with the inclusion of measurement year in the logistic regression. However highlighting the odds of a specific type of ER visit increasing year over year has important implications for how resources should be prioritized in ER settings.

Please note, if each case is an independent ER visit then authors are measuring if the proportions of ER visits for one disease are different year to year. ER visits that are "not" the disease being measured have no chance of being the disease being measured because they are independent observations. This is different from describing if there was a noticeable increase or decrease in ER visits for a specific disease. For example, saying the odds of one disease state increasing compared to all ER visits means the proportion of ER visits for one disease relative to others is higher in the given year. But it does not tell us if the actual rate of cases for that specific ED visit changed from year over year. If authors wish to not modify the analysis this should at least be noted in the Limitations. Year to year changes in proportions are of course associated with the change in the incidence of an event, but they are not exactly the same unless there is a guarantee that cases that are not the event in question are constant year to year.

The example the authors provided is different from the authors own analysis. In the study authors recommended they are using a logistic regression to determine if the measurement years influenced the the use (yes or no, a binary variable) of a specific healthcare provider. Each individual case is a patient. Assessing the influence of measurement year is to see if the year they are asked influenced their response. Each patient has a chance to use or not use the healthcare provider in question in the specific measurement year.

Author Response

Dear Reviewer,

We appreciate your insightful comment. Please find the response below:

Reviewer 1:

  1. Thank you for addressing my concerns in the revision. I appreciate the work the reviewers have done. I still do not agree with the inclusion of measurement year in the logistic regression. However highlighting the odds of a specific type of ER visit increasing year over year has important implications for how resources should be prioritized in ER settings. Please note, if each case is an independent ER visit then authors are measuring if the proportions of ER visits for one disease are different year to year. ER visits that are "not" the disease being measured have no chance of being the disease being measured because they are independent observations. This is different from describing if there was a noticeable increase or decrease in ER visits for a specific disease. For example, saying the odds of one disease state increasing compared to all ER visits means the proportion of ER visits for one disease relative to others is higher in the given year. But it does not tell us if the actual rate of cases for that specific ED visit changed from year over year. If authors wish to not modify the analysis this should at least be noted in the Limitations. Year to year changes in proportions are of course associated with the change in the incidence of an event, but they are not exactly the same unless there is a guarantee that cases that are not the event in question are constant year to year. The example the authors provided is different from the authors own analysis. In the study authors recommended they are using a logistic regression to determine if the measurement years influenced the the use (yes or no, a binary variable) of a specific healthcare provider. Each individual case is a patient. Assessing the influence of measurement year is to see if the year they are asked influenced their response. Each patient has a chance to use or not use the healthcare provider in question in the specific measurement year.

Reply: Thank you for the comment. The rate and percentage of our variables have been indicated in Table 1. The following sentence was added to the limitation (highlighted in the text):

We analyzed the odds of certain conditions over the year. The increase in odds indicates that the proportion of ED visits for one condition versus others is higher in a given year, but it does not necessarily imply that the number of that specific ED visit has increased over the year. In our analysis, however, the observed increases in odds were often accompanied by corresponding increases in numbers (Table 1).